



# A Method for Preliminary Rotor Design - Part 2:
# Wind Turbine Rotor Optimization with Radial Independence

Kenneth Loenbaek[1,2], Christian Bak[2], and Michael McWilliam[2]

[1]Suzlon Blade Science Center, Brendstrupgaardsvej 13, 8210 Aarhus, Denmark
[2]Technical University of Denmark, Frederiksborgvej 399, 4000 Roskilde, Denmark

**Correspondence:** Kenneth Loenbaek (kenneth.loenbaek@suzlon.com)

**Abstract.** A novel wind turbine rotor optimization methodology is presented. Using an assumption of radial independence it is possible to obtain an optimal relationship between the global power- ($C_P$) and load-coefficient ($C_T$, $C_{FM}$) through the use of KKT-multipliers, leaving an optimization problem that can be solved at each radial station independently. It allows to solve load constraint power and Annual-Energy-Production (AEP) optimization problems where the optimization variables are only the KKT-multipliers (scalars), one for each of the constraint. For the paper two constraints, namely the thrust and blade-root-flap-moment is used, leading to two optimization variables.

Applying the optimization methodology to maximize power ($P$) or Annual-Energy-Production (AEP) for a given thrust and blade-root-flap-moment, but without a cost-function, leads to the same overall result with the global optimum being unbounded in terms of rotor radius ($\tilde{R}$) with a global optimum being at $\tilde{R} \to \infty$. The increase in power and AEP is in this case $\Delta P = 50\%$ and $\Delta AEP = 70\%$, with a baseline being the Betz-optimum rotor.

With a simple cost function and with the same setup of the problem a Power-per-Cost (PpC) optimization resulted in a Power-per-Cost increase of $\Delta PpC = 4.2\%$ with a radius increase of $\Delta R = 7.9\%$ as well as a power increase of $\Delta P = 9.1\%$. This was obtained while keeping the same flap-moment and reaching a lower thrust of $\Delta T = -3.8\%$. The equivalent for AEP-per-Cost (AEPpC) optimization leads to increased cost-efficiency of $\Delta AEPpC = 2.9\%$ with a radius increase of $\Delta R = 17\%$ and an AEP increase of $\Delta AEP = 13\%$, again with the same, maximum flap-moment, while the maximum thrust is $-9.0\%$ lower than the baseline.

**Keyword:** Preliminary rotor design, initial rotor design

## 1 Introduction

Wind turbine design optimization has been an integral part of wind turbine design since the start of the wind turbine industry. The target for such optimization has varied greatly from pure aerodynamic optimization with the target to maximize the power extraction (see: Manwell et al. (2010), Sørensen (2016), Jamieson (2018)), to a more holistic turbine design where the target is to minimize the cost of the turbine through modeling the physics of the turbine components as well as their associated, see e.g. Fuglsang et al. (2002), Hjort et al. (2009), Bottasso et al. (2010), Dykes and Meadows (2012), Perez-Moreno et al. (2016). Common to these approaches is connecting a set of simulations tools (e.g. BEM-solver, structural-solver, controller, etc.)



through a cost-function, leading to a fairly complicated optimization problem, with a lot of design variables. As a consequence, the computational time for each evaluation of the objective function might be infeasible for exploring the design space and carrying out sensitivity studies considering the number of design variables. Exploring the design space is especially important for the preliminary design phase where e.g. the rotor size and rated power, need to be determined.

Lately, some research has been made within preliminary rotor design which seems to have started with the concept of Low-Induction-Rotors (Chaviaropoulos and Voutsinas, 2012) where they investigate the optimal constant axial-induction ($a$) with a flap-moment constraint, arriving at an optimum of $a = 0.2$. A similar study was made by Buck and Garvey (2015b) where they used a cost-function to find the most cost-effective rotor to have $a = 0.25$. They also made a study (Buck and Garvey, 2015a) where they investigated so-called thrust-clipping (limiting the maximum thrust) as a means to find the optimal cost-effective rotor. This author recently made a study (Loenbaek et al., 2020a) where the approach taken by Chaviaropoulos Chaviaropoulos and Voutsinas (2012) was generalized to include additional constraints (e.g. tip-deflection as well as constant mass). This study investigated the impact on the power-curve, where thrust-clipping is found to be the design concept that leads to the largest energy increase, as compared to the Low-Induction-Rotor design concept.

Common to these studies is the assumption of constant axial-induction along the rotor span. There have also been some studies to investigate the impact of allowing the axial-induction to change along the rotor span. Kelley (2017) investigates the optimal distribution of $a$, showing that keeping a fixed maximum bending moment the optimal $a$-distribution is tapering towards the tip of the blade. Recently a study by Jamieson (2020) extended the work of Chaviaropoulos and Voutsinas (2012) where they allow for variations in $a$ along the span, showing that it is possible to reach the same power increase, but with a smaller radius increase. They also see a similar tapering $a$-distribution towards the tip as Kelley (2017). The current study builds on top of this works and it could be seen as an extension of previous work by this author (Loenbaek et al., 2020a), where a variation in $a$ (or loading) along the rotor span is added, as well as including a simple cost-function. The developed optimization methodology described in this paper is part 2 of 2 part paper, where part 1 (Loenbaek et al., 2020b) describes the aerodynamic model used thought out this paper.

In this paper, an optimization methodology is presented which aims to maximize the Power ($P$) or Annual-Energy-Production (AEP) with a fixed radius increase. Since the pure aerodynamic optimization leads to an unbounded optimum a simple cost-function is introduced leading to Power-per-Cost (PpC) and AEP-per-Cost optimization. The aerodynamic- and cost-modeling is kept at a fairly simple level with BEM-like aerodynamics and simple radius dependent cost-functions. It allows for the optimization problem to be solved for the global optimum within numerical accuracy. The crucial assumption made for this to be possible is the assumption of radial independence which allows the optimization problem to be made into a set of nested optimizations, each resulting in a well-behaved optimization problem. A key innovation is that the optimization is based on loading, not the design variables (e.g. control points for chord and twist), which leads to a large reduction in the number of design variables and a simplification of the optimization problem. Thus, in contrast to many methods used to optimize wind turbine rotors, this method is very simple. Even though it is simple it is thought to be an important step for preliminary rotor design where one would like to investigate the impact of changes in the cost-function or constraints. This is especially important where technology improvements should be targeted in order to lead to the biggest improvements in PpC or AEPpC.





This paper is split into 2 sections, starting with the *Optimization Methodology* where the optimization problem is presented and the process of solving the optimization problem with the assumption of radial independence is then given. Then a *Results and discussion* section, where the results from solving the optimization problem are presented and discussed.

## 2   Optimization Methodology

In this section, we will present an optimization methodology for wind turbine rotor optimization. It is named *Wind turbine*
*rotor Optimization with Radially Independence* (WOwRI). Before presenting WOwRI a discussion of the assumptions as well as the terminology is given ending with a short discussion of the aerodynamic solver used. Then WOwRI is presented for power optimization with a fixed radius increase as well as wind speed. WOwRI is then extended for AEP-optimization with a fixed radius increase and at last, WOwRI is extended for optimization with a simple cost-function to determine optimal rotor size.

The core assumption for WOwRI is the assumption of radial independence and to explaining what it means the distinction between *global-* and *local*-rotor-variables is introduced. Global-rotor-variables is a scalar value for the whole rotor (e.g. power, thrust, ..) whereas local-rotor-variables is a scalar at a given rotor-radius ($r$) location (e.g. lift, drag, ..). With this definition, the assumption of radial independence is applied for the local-rotor-variables meaning that changes in the loading (like lift) at one radial location will not affect the flow state (flow through the rotor plane) at any other radial location. This is the same
assumption made for Blade-Element-Momentum theory (Sørensen, 2016, p. 99).

An assumption that is related to the radial independence is a direct relationship between the local thrust loading and the local power at the same radial location. It means that if the local thrust loading is given the local power can be computed. This is further discussed if section 2.1.

Throughout this paper, the flow is assumed to be steady-state. As a consequence, when optimization is made with load
constraints (e.g. thrust and flap-moment) it is the steady state load that is constrained. But for current utility scale wind turbine design it is common that the design is driven by the dynamic extreme loads. It means that the underlying assumption for this optimization methodology is that a constraint steady state load is in some way connected with the dynamic extreme load. This assumption is however not tested in this paper.

WOwRI is based on power ($P$) optimization with a given set of load constraints. These constraints can be (but not limited
to): thrust, flap-moment, tip-deflection, and max stress/strain. Where the key requirement for the constraint to be suited for WOwRI is that it satisfies the radial independence requirement. A form that satisfied (but not limited to) this requirement is:

$$X_{con} = \int\limits_0^R \frac{\partial T}{\partial r}(r) \cdot f_X(r) dr \tag{1}$$

Where $X_{con}$ is a global-rotor-variable (like thrust, $T$), $\frac{\partial T}{\partial r}(r)$ is the thrust loading density (loading per meter), $f_X(r)$ is a function that change the impact of thrust-loading-density as each radial station. This is a rather abstract definition, but showing
how an extensive list of constraints is related to this definition is though to be out of scope for this paper since the purpose





is to present the optimization methodology. Instead, the focus will be on two specific constraints, namely a thrust ($T$) and blade-root-flap-bending-moment ($M_f$) constraints. These two constraints are given as:

$$T = \int_0^R \frac{\partial T}{\partial r} dr \qquad\qquad \text{(thrust constraint, with: } f_X = 1) \qquad (2)$$

$$M_f = \int_0^R \frac{\partial T}{\partial r} r dr \qquad\qquad \text{(flap-moment constraint, with: } f_X = r) \qquad (3)$$

Where the relationship with the generalized constraint form shown in equation 1 is given in the parenthesis.

## 2.1 The Aerodynamic solver

The aerodynamic solver (Radially Independent Actuator Disc model - RIAD) used though out this paper is further described in *Part I* and a brief overview is therefore only given here. It makes an explicit relationship between the *local-thrust-coefficient* ($C_{LT}$ - normalized $\partial T/\partial r$) and the *local-power-coefficient* ($C_{LP}$ - normalized $\partial P/\partial r$) with given operational conditions 100 such as the global tip-speed-ratio ($\lambda$), the local glide-ratio ($C_l/C_d$) and may include tip-loss as well. A sketch showing the relationship graphically can be seen in figure 1

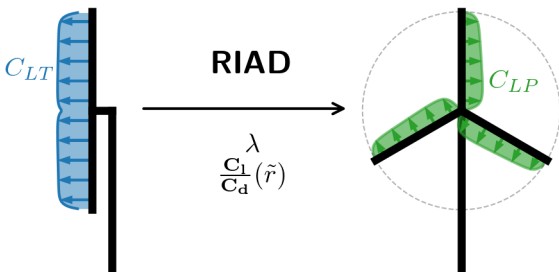

**Figure 1.** Sketch showing a diagram for the Radially Independent Actuator Disc model (RIAD).

## 2.2 Power optimization

In this section, the optimization methodology that allows for the fast and very efficient solution to the optimization is derived. It finds the optimal power for a fixed rotor increase. In principle, the rotor radius could as well be an optimization parameter 105 but as it is shown later, the optimal global power turns out to be unbounded, and having the solution for fixed rotor radius increase allows for optimization with a simple cost-function, which is further explained later. The main outcome of this section is a function, that through solving an optimization problem gives the optimal power for a given set of constraints with a fixed radius increase and fixed wind speed ($P_{opt}(R, V)$).



### 2.2.1 Problem formulation

The optimization problem is maximizing power ($P$) with two constraints, the maximum allowable thrust ($T_0$) and a maximum allowable blade-root-flap-bending moment ($M_f$) for a fixed rotor radius and fixed wind speed. The design variable is the distributed thrust loading along the span of the rotor $\left(\frac{\partial \boldsymbol{T}}{\partial \boldsymbol{r}}(r)\right)$. It is important to notice that the distributed load is a function of r or when discretized a vector.

Mathematically the problem can be stated as:

$$\max_{\frac{\partial \boldsymbol{T}}{\partial \boldsymbol{r}}} P\left(\frac{\partial \boldsymbol{T}}{\partial \boldsymbol{r}}\right) \tag{4}$$

$$\text{subj.} \quad \begin{aligned} T\left(\frac{\partial \boldsymbol{T}}{\partial \boldsymbol{r}}\right) &\leq T_0 \\ M_f\left(\frac{\partial \boldsymbol{T}}{\partial \boldsymbol{r}}\right) &\leq M_{f,0} \end{aligned}$$

Where the bold-face $\frac{\partial \boldsymbol{T}}{\partial \boldsymbol{r}}$ signifies that it is a function and not just a scalar. The zero-subscript is denoting a constraint limit.

### 2.2.2 Problem formulation in integral form and normalization

Using the same normalization as in Part 1 (Loenbaek et al., 2020b, sec. 2.1 eq. 3-6 ) the power and constraints can be normalized

as:

$$\tilde{P} = \tilde{R}^2 \tilde{V}^3 C_P = \tilde{R}^2 \tilde{V}^3 2 \int_0^1 C_{LP}(C_{LT}(\tilde{r})) \cdot \tilde{r} d\tilde{r} \tag{5}$$

$$\tilde{T} = \tilde{R}^2 \tilde{V}^2 C_T = \tilde{R}^2 \tilde{V}^2 2 \int_0^1 C_{LT}(\tilde{r}) \cdot \tilde{r} d\tilde{r} \tag{6}$$

$$\tilde{M}_f = \tilde{R}^3 \tilde{V}^2 C_{FM} = \tilde{R}^3 \tilde{V}^2 3 \int_0^1 C_{LT}(\tilde{r}) \cdot \tilde{r}^2 d\tilde{r} \tag{7}$$

Where $\tilde{R} = \frac{R}{R_0}$ and $\tilde{V} = V/V_{rated,0}$, with $R_0$ being a reference radius and $V_{rated,0}$ the rated wind speed for a reference turbine.

Both are related to the constraint limit. The optimization problem can therefore be reformulated as:

$$\max_{\boldsymbol{C_{LT}}} C_P\left(\boldsymbol{C_{LT}}\right) \cdot \tilde{R}^2 \tilde{V}^3 \tag{8}$$

$$\text{subj.} \quad \begin{aligned} C_T\left(\boldsymbol{C_{LT}}\right) \cdot \tilde{R}^2 \tilde{V}^2 &\leq \tilde{T}_0 \\ C_{FM}\left(\boldsymbol{C_{LT}}\right) \cdot \tilde{R}^3 \tilde{V}^2 &\leq \tilde{M}_{f,0} \end{aligned}$$

### 2.2.3 Reformulating as a Lagrange objective function

The optimization problem stated in the previous sections has a solution that need to satisfy the *Karush-Kuhn-Tucker* (KKT)

(Kuhn and Tucker, 1951) theorem to be optimal. It means that a solution to the original problem can also be found by solving the optimization problem in Eq. (9), where the objective function has been reformulated as a Lagrange objective function ($\mathcal{L}^*$)





(including the constraints in the objective function):

$$\max_{\boldsymbol{C_{LT}}, W_0^*, W_1^*} \mathcal{L}^* = \max_{\boldsymbol{C_{LT}}, W_0^*, W_1^*} \left[ C_P\left(\boldsymbol{C_{LT}}\right) \cdot \tilde{R}^2 \tilde{V}^3 - W_0^* \left[ C_T\left(\boldsymbol{C_{LT}}\right) \cdot \tilde{R}^2 \tilde{V}^2 - \tilde{T}_0 \right] - W_1^* \left[ C_{FM}\left(\boldsymbol{C_{LT}}\right) \cdot \tilde{R}^3 \tilde{V}^2 - \tilde{M}_{f,0} \right] \right]$$
(9)

Where $W_i^*$ are the so called KKT-multipliers with the property $W_i^* \geq 0$. These $W_i^*$ needs to be adjusted for active constraints until the constraint is meet. For an inactive constraint $W_i^* = 0$.

The key point for rewriting the optimization as a Lagrange objective function is to be able to solve the optimization of the $\boldsymbol{C_{LT}}$-distribution. To do this we will look at the case where $W_0^*, W_1^*$ is constant input parameters. Since the location of the optimum dose not change with scaling and a constant offset, a new Lagrange objective function can be written as:

$$\max_{\boldsymbol{C_{LT}}} \mathcal{L} = \max_{\boldsymbol{C_{LT}}} \left[ C_P\left(\boldsymbol{C_{LT}}\right) - W_0 C_T\left(\boldsymbol{C_{LT}}\right) - W_1 C_{FM}\left(\boldsymbol{C_{LT}}\right) \right]$$
(10)

Where scaling in front of $C_T$ and $C_{FM}$ has be absorbed into $W_0$ and $W_1$ respectively (notice the change from $W_i^*$ to $W_i$ to stress that they has been rescaled between equation 9 and 10). Any solution for the Lagrange function 10 (in terms of $\boldsymbol{C_{LT}}$) will also be a solution to Lagrange function 9 and here by a solution for the optimization problem 8, for some set of constraint limits. But which set of constraint is not known a-priori to solving the optimization problem. Equation 10 is also sometimes referred to as the Pareto-optimal problem for $C_P, C_T, C_{FM}$, giving the maximum $C_P$ for a given value of $C_T, C_{FM}$ or any combination of the two. By varying the $W_i$'s the location on the so-called Pareto-optimal surface is changed.

### 2.2.4 Solving for the optimal loading distribution

In this section we will apply the assumption of radial independence to show that the optimal solution for the trade-off between global power ($C_P$) and the loading ($C_T, C_{FM}$) can be found for each radial station independently. In integral form the optimization for the optimal loading reads:

$$\max_{\boldsymbol{C_{LT}}} \mathcal{L} = \max_{\boldsymbol{C_{LT}}} \left[ 2 \int_0^1 C_{LP}\left(C_{LT}(r)\right) \cdot \tilde{r} d\tilde{r} - W_0 2 \int_0^1 C_{LT}(r) \cdot \tilde{r} d\tilde{r} - W_1 3 \int_0^1 C_{LT}(r) \cdot \tilde{r}^2 d\tilde{r} \right]$$
(11)

The 3 integration's can be combined into one since $W_i$ is independent on $\tilde{r}$. Then applying the radial independence the maximization can be moved within the integration:

$$\max_{\boldsymbol{C_{LT}}} \mathcal{L} = \int_0^1 \max_{C_{LT}} \left[ 2 C_{LP}(C_{LT}) \tilde{r} - 2 W_0 C_{LT} \tilde{r} - 3 W_1 C_{LT} \tilde{r}^2 \right] d\tilde{r}$$
(12)

The step between optimization problem 11 and 12 transform the optimization problem from a problem of finding a distribution for $C_{LT}$ to a problem of finding a scalar value for $C_{LT}$ at each radial station ($\tilde{r}$), which is a significant simplification of the problem. This is also signified by the drop of the bold-face $C_{LT}$.

Introducing the local Lagrange objective function ($\mathcal{L}_L$) the optimization problem at each radial station can be formulated as:

$$\max_{C_{LT}} \mathcal{L}_L = \max_{C_{LT}} \left[ 2 C_{LP}(C_{LT}) \tilde{r} - 2 W_0 C_{LT} \tilde{r} - 3 W_1 C_{LT} \tilde{r}^2 \right]$$
(13)





To solve this problem it is assumed that $C_{LP}$ is a well behaved function, like the function presented in Part 1, which means
that the problem can be solved as:

$$\max_{C_{LT}} \mathcal{L}_L \implies \frac{\partial \mathcal{L}_L}{\partial C_{LT}} = 2\tilde{r}\frac{\partial C_{LP}}{\partial C_{LT}} - 2W_0\tilde{r} - 3W_1\tilde{r}^2 = 0 \qquad\qquad C_{LT} \in \left[-\frac{8}{9}, \frac{8}{9}\right] \qquad (14)$$

Where $-\frac{8}{9}$ lower limit is an arbitrary lower limit. By using $\frac{\partial C_{LP}}{\partial C_{LT}}$ from *Part 1, eq. XX* the optimization problem can be reduced
to a root finding problem, which can be solved though the use of a root finding algorithms like bisection or Brent's method.
From now on it is assumed that the solution for the optimization problem in 14 can be solved for any level of resolution in $\tilde{r}$
for a given input of $W_0, W_1$. It hereby makes a function that takes $W_0, W_1$ as inputs and returns the optimal $C_{LT}$-distribution,
denoted by $C_{LT,opt}$. As it was mentioned before these $C_{LT}$-distributions will also be a solutions to the original problem as
presented in 8 for a set of constraint limits. A flow-chart showing how $C_{LT,opt}$ is found for a given set of inputs can be seen in
figure 2.

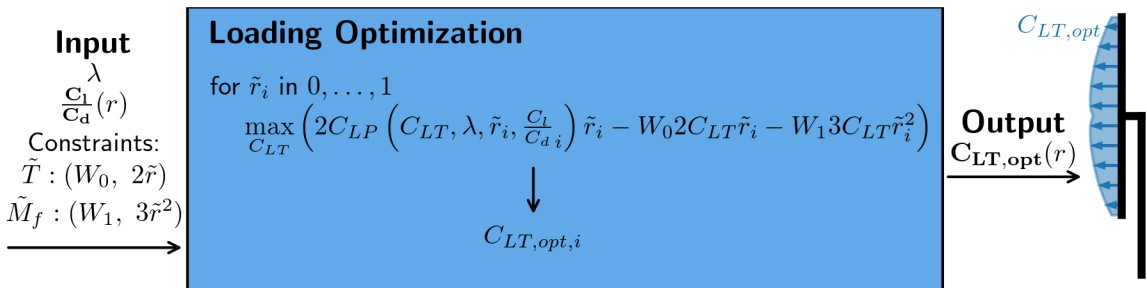

**Figure 2.** Flow-chart for the Loading Optimization with a given set of inputs. (Aerodynamic input: $\lambda, C_l/C_d$ and constraints input: $\tilde{T} \leq \tilde{T}_0 \rightarrow W_0, f_X = 1, \tilde{M}_f \leq \tilde{M}_{f,0} \rightarrow W_1, f_X = \tilde{r}$)

### 2.2.5 The optimization problem with a function for optimal loading

With the $C_{LT,opt}$ function mapping the input $W_0, W_1$ to an optimal $C_{LT}$-distribution the optimization problem presented in 8
can be changed from an optimization for a distribution ($C_{LT}$) to an optimization in 2 scalars ($W_0, W_1$) which is a significant
simplification of the original problem:

$$\max_{W_0, W_1} \quad C_P(W_0, W_1) \cdot \tilde{R}^2 \tilde{V}^3 \qquad\qquad\qquad (15)$$

$$\text{subj.} \quad \begin{aligned} C_T(W_0, W_1) \cdot \tilde{R}^2 \tilde{V}^2 &\leq \tilde{T}_0 \\ C_{FM}(W_0, W_1) \cdot \tilde{R}^3 \tilde{V}^2 &\leq \tilde{M}_{f,0} \end{aligned}$$

$$\Downarrow$$

$$\tilde{P}_{opt}(\tilde{V}, \tilde{R}) \qquad\qquad\qquad (16)$$

The optimization problem can be solved with most optimization algorithms capable of solving constraint optimization prob-
lems. All the optimization problems solved in this paper is solved with the use of the Python Scipy optimizer (Virtanen et al.,





2020). A flow-chart showing the optimization process can be seen in figure 3. Notice that it dependence on the *Loading Op-*

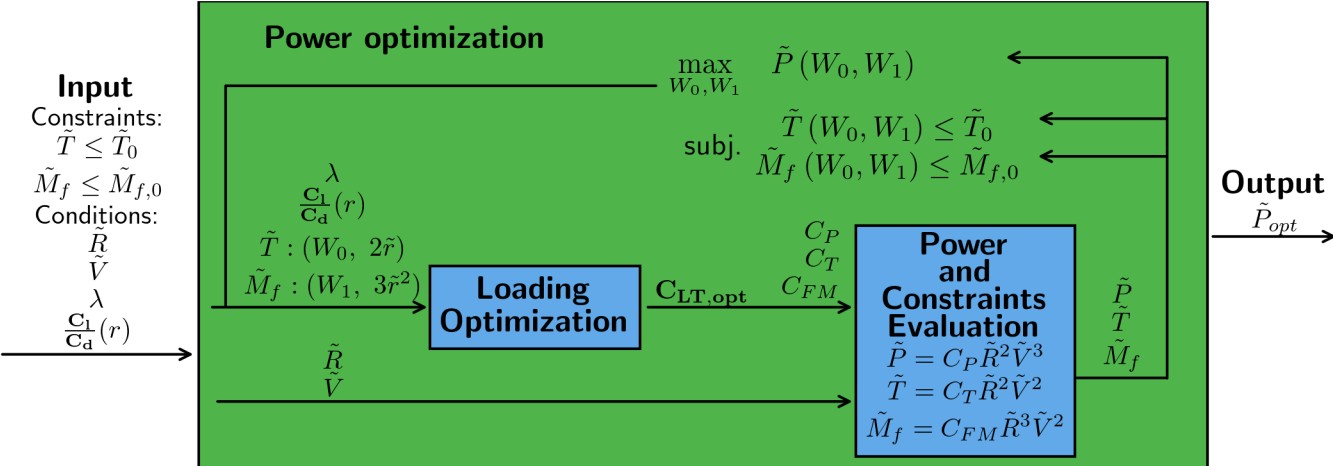

**Figure 3.** Flow-chart for *Power Optimization*. Notice that the *Loading Optimization* is nested within the optimization loop. The optimizer needs to adjust the $W_0, W_1$ for maximum power while the constraints are satisfied.

*timization*, meaning that this is a nested optimization loop. The output from the optimization is the optimal power ($\tilde{P}_{opt}$) that satisfy the constraints for a fixed rotor increase ($\tilde{R}$) and fixed wind speed ($\tilde{V}$).

### 2.3 AEP optimization

The purpose of this section is to extend the optimization methodology to include optimization for maximum *Annual Energy Production* (AEP) with load constraints across all wind speeds as well as fixed rated power and a fixed radius increase.

AEP is computed as the average power over a year multiplied by the time of a year. The average power can be computed from the wind distribution ($f_{wei}$, i.e. the frequency that a wind turbine is operating at a given wind speed) and the power-curve. Mathematically it can be computed as:

$$AEP = T_{year} \int_{V_{CI}}^{V_{CO}} P\left(V, \frac{\partial \boldsymbol{T}}{\partial \boldsymbol{r}}(V), R\right) f_{wei} dV \tag{17}$$

Where $T_{year}$ is the time of a year, $P$ the power-curve function, $V_{CI}$, and $V_{CO}$ it the Cut-In- and Cut-Out-wind-speed respectively. The optimization problem for AEP optimization can be stated as:

$$\max_{\frac{\partial \boldsymbol{T}}{\partial \boldsymbol{r}}(V)} T_{year} \int_{V_{CI}}^{V_{CO}} P\left(V, \frac{\partial \boldsymbol{T}}{\partial \boldsymbol{r}}(V)\right) \cdot f_{wei} dV \tag{18}$$

$$\text{subj.} \quad \left.\begin{array}{rl} T\left(V, \frac{\partial \boldsymbol{T}}{\partial \boldsymbol{r}}(V)\right) & \leq T_0 \\ M_f\left(V, \frac{\partial \boldsymbol{T}}{\partial \boldsymbol{r}}(V)\right) & \leq M_{f,0} \\ P\left(V, \frac{\partial \boldsymbol{T}}{\partial \boldsymbol{r}}(V)\right) & \leq P_{rated} \end{array}\right\} \text{for all } V$$





Where it should be noticed that the loading is allowed to change freely with changing wind speed, which is indicated by the $\frac{\partial \boldsymbol{T}}{\partial \boldsymbol{r}}(V)$. Applying the same normalization as in section 2.2.2, where the normalization of the wind speed is the rated wind speed for the rotor with $\tilde{R} = 1$. The normalized optimization problem is given as:

$$\max_{\boldsymbol{C_{LT}}(\tilde{V})} \int_{\tilde{V}_{CI}}^{\tilde{V}_{CO}} C_P\left(\boldsymbol{C_{LT}}(\tilde{V})\right) \cdot \tilde{R}^2 \tilde{V}^3 f_{wei} d\tilde{V} \tag{19}$$

$$\text{subj.} \quad \left. \begin{array}{rl} C_T\left(\boldsymbol{C_{LT}}(\tilde{V})\right) \cdot \tilde{R}^2 \tilde{V}^2 & \leq \tilde{T}_0 \\ C_{FM}\left(\boldsymbol{C_{LT}}(\tilde{V})\right) \cdot \tilde{R}^3 \tilde{V}^2 & \leq \tilde{M}_{f,0} \\ C_P\left(\boldsymbol{C_{LT}}(\tilde{V})\right) \cdot \tilde{R}^2 \tilde{V}^3 & \leq \tilde{P}_0 \end{array} \right\} \text{ for all } \tilde{V}$$

Using the assumption that $\boldsymbol{C_{LT}}$ can change independently with wind speed the maximization can be taken within the wind speed integration. Since the constraints is for all wind speeds, the optimization problem is now a *Power Optimization* for each wind speed.

$$\int_{\tilde{V}_{CI}}^{\tilde{V}_{CO}} \max_{\boldsymbol{C_{LT}}} \left[ C_P\left(\boldsymbol{C_{LT}}\right) \cdot \tilde{R}^2 \tilde{V}^3 \right] f_{wei} d\tilde{V} \tag{20}$$

$$\text{subj.} \quad \left. \begin{array}{rl} C_T\left(\boldsymbol{C_{LT}}(\tilde{V})\right) \cdot \tilde{R}^2 \tilde{V}^2 & \leq \tilde{T}_0 \\ C_{FM}\left(\boldsymbol{C_{LT}}(\tilde{V})\right) \cdot \tilde{R}^3 \tilde{V}^2 & \leq \tilde{M}_{f,0} \\ C_P\left(\boldsymbol{C_{LT}}(\tilde{V})\right) \cdot \tilde{R}^2 \tilde{V}^3 & \leq \tilde{P}_0 \end{array} \right\} \text{ for all } \tilde{V}$$

$$\Downarrow \text{ (Power Optimization)}$$

$$\int_{\tilde{V}_{CI}}^{\tilde{V}_{CO}} \max_{\boldsymbol{W_0}, \boldsymbol{W_1}} \left[ C_P\left(\boldsymbol{W_0}, \boldsymbol{W_1}\right) \cdot \tilde{R}^2 \tilde{V}^3 \right] f_{wei} d\tilde{V} \tag{21}$$

$$\text{subj.} \quad \left. \begin{array}{rl} C_T\left(\boldsymbol{W_0}, \boldsymbol{W_1}\right) \cdot \tilde{R}^2 \tilde{V}^2 & \leq \tilde{T}_0 \\ C_{FM}\left(\boldsymbol{W_0}, \boldsymbol{W_1}\right) \cdot \tilde{R}^3 \tilde{V}^2 & \leq \tilde{M}_{f,0} \\ C_P\left(\boldsymbol{W_0}, \boldsymbol{W_1}\right) \cdot \tilde{R}^2 \tilde{V}^3 & \leq \tilde{P}_0 \end{array} \right\} \text{ for all } \tilde{V}$$

Where the bold-face $\boldsymbol{W_0}, \boldsymbol{W_1}$ signifies that it is changing with wind speed.

It can be further simplified as:

$$A\tilde{E}P_{opt}(\tilde{R}) = \int_{\tilde{V}_{CI}}^{\tilde{V}_{CO}} \tilde{P}_{opt}(\tilde{V}, \tilde{R}) f_{wei} d\tilde{V} \tag{22}$$

Where the function for the output from the *Power Optimization* ($\tilde{P}_{opt}$) is used. It shows that the *AEP Optimization* can be reduced to a Power Optimization for each wind speed in the integration. A flow-chart for the AEP Optimization can be seen in figure 4. The output from the optimization is denoted as $A\tilde{E}P_{opt}$.



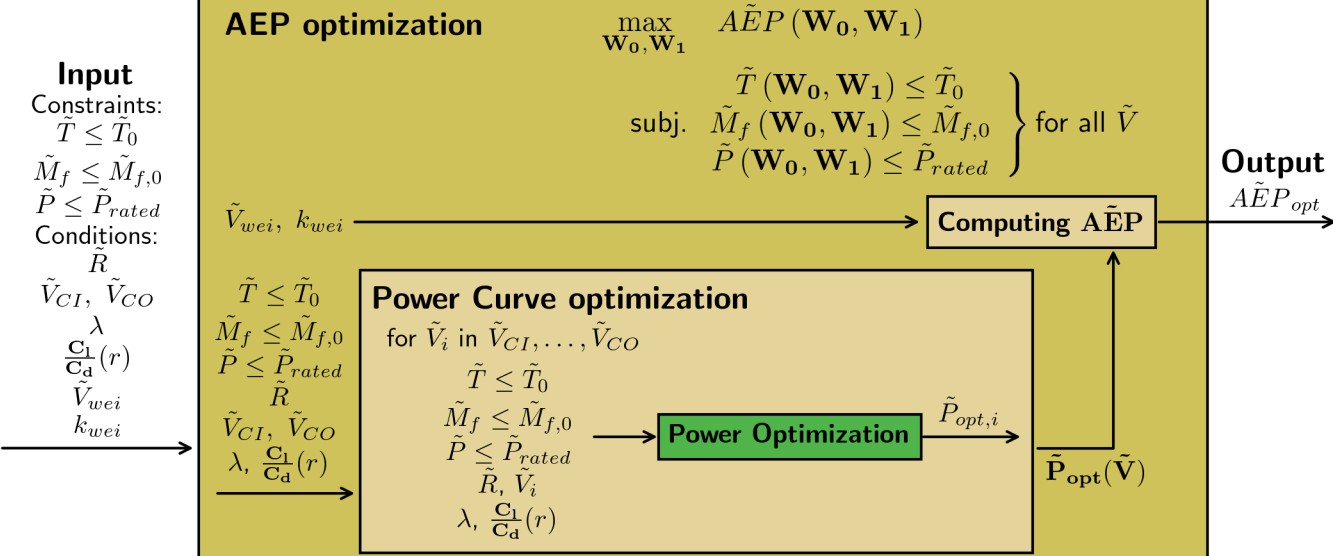

**Figure 4.** Flow-chart for the *AEP Optimization*. The optimization is simply a *Power Optimization* for each wind speed in the power curve.

## 2.4 WOwRI optimization with a simple cost function

The optimizations presented so far have been for a fixed radius increase, but in this section, the optimization for rotor radius will be presented. The *Power Optimization* and *AEP Optimization* could in principle easily be extended for radius optimization as well by simply adding the rotor radius as a design variable but as it is discussed in section 3.1 the optimization problem is unbounded with the global optimum at $\tilde{R} \to \infty$ which is clearly not feasible for turbine design. To get a feasible rotor design, the optimization for rotor size will also include a cost function.

### 2.4.1 Cost function

The current work focuses on preliminary wind turbine rotor design and detailed cost function like the one in Fingersh et al. (2006) is therefore thought to be out of scope. A simple cost function that is purely a function of the rotor radius is therefore proposed here.

The cost function will roughly estimate the mass increase associated with the increase in rotor radius, with the underlying assumption that mass and cost scales roughly in the same way. It is important to notice here that it is not the whole turbine and associated components that need to be scaled with the change in rotor radius, as the optimization is a load constrained optimization, the loads do not change and the associated components, therefore, do not need to be scaled.

The cost model is simply based on a *cost-fraction* which is the fraction of the cost that is affected by changes in radius, as well as the *cost-exponent* which describes how the cost (or mass) for this cost-fraction scale with changes in radius. If the components affected by the radius increase are assumed to be the blades, tower, and foundation the cost-fraction is found



to 39% (using the number from (Stehly and Beiter, 2020, p.7 fig 1)). The cost-exponent is bound in the range $1-3$, as an exponent of 3 would be for the case where the mass increase in all 3 dimensions, whereas 1 is the case where the mass is only increaseing in 1 dimension (e.g. tip-extension). With the load constraints, it is definitely less than 3 and a good estimate for the cost-exponent is therefore thought to be 1.5. The suggested normalized cost function is given as:

$$\tilde{f}_{cost}(\tilde{R}) = 0.39 \cdot \tilde{R}^{1.5} + 0.61 \tag{23}$$

It is important to notice that this is a rough estimate for a cost function and more importantly it has a great impact on the optimal rotor radius. But as the purpose of this paper is to present the WOwRI optimization methodology it is thought to be out of scope for this paper to investigate it further here.

### 2.4.2 Rotor size optimization with cost function

The outcome from section 2.2 and 2.3 was the functions $\tilde{P}_{opt}(\tilde{R})$ (assuming $\tilde{V} = 1$) and $A\tilde{E}P_{opt}(\tilde{R})$ respectively. These functions computes the optimal power/AEP for a given set of constraints at a fixed radius increase. Using these functions the following optimization problems for the optimal radius increase can stated as:

$$\max_{\tilde{R}} \frac{\tilde{P}_{opt}(\tilde{R})}{f_{cost}(\tilde{R})} \qquad (\textit{Power-per-Cost Optimization}) \tag{24}$$

$$\max_{\tilde{R}} \frac{A\tilde{E}P_{opt}(\tilde{R})}{f_{cost}(\tilde{R})} \qquad (\textit{AEP-per-Cost Optimization}) \tag{25}$$

Where the impact of the constraints on the optimal design is implicitly captured in $\tilde{P}_{opt}$ and $A\tilde{E}P_{opt}$.



## 3 Results and discussion

245 In this section, the result of applying the WOwRI optimization methodology is presented. At first, the result of pure power optimization at a single wind speed is presented and discussed, then the result of including a cost-function for so-called Power-per-Cost optimization which leads to a turbine blade planform design is presented and discussed. AEP-optimization is then presented and then at the end AEP-per-Cost optimization is presented which leads to the optimal power-curve. At the very end, it is tested how close it is possible to get to the optimal power-curve with common wind turbine technology.

### 3.1 Power optimization

250

This section shows the result of applying the optimization methodology described in section 2.2 for increasing the rotor radius.

The input for the aerodynamic solver (Part 1, Loenbaek et al. (2020b)) is as simple as possible with no viscous loss ($C_d/C_l = 0$) and without tip-loss for two different tip-speed-ratios ($\lambda \to \infty$, $\lambda = 5$).

In figure 5 the optimization problem is solved for increasing values of $\tilde{R}$ and the power is relative to the baseline power ($\tilde{P}_0$) which is the power at $\tilde{R} = 1$.

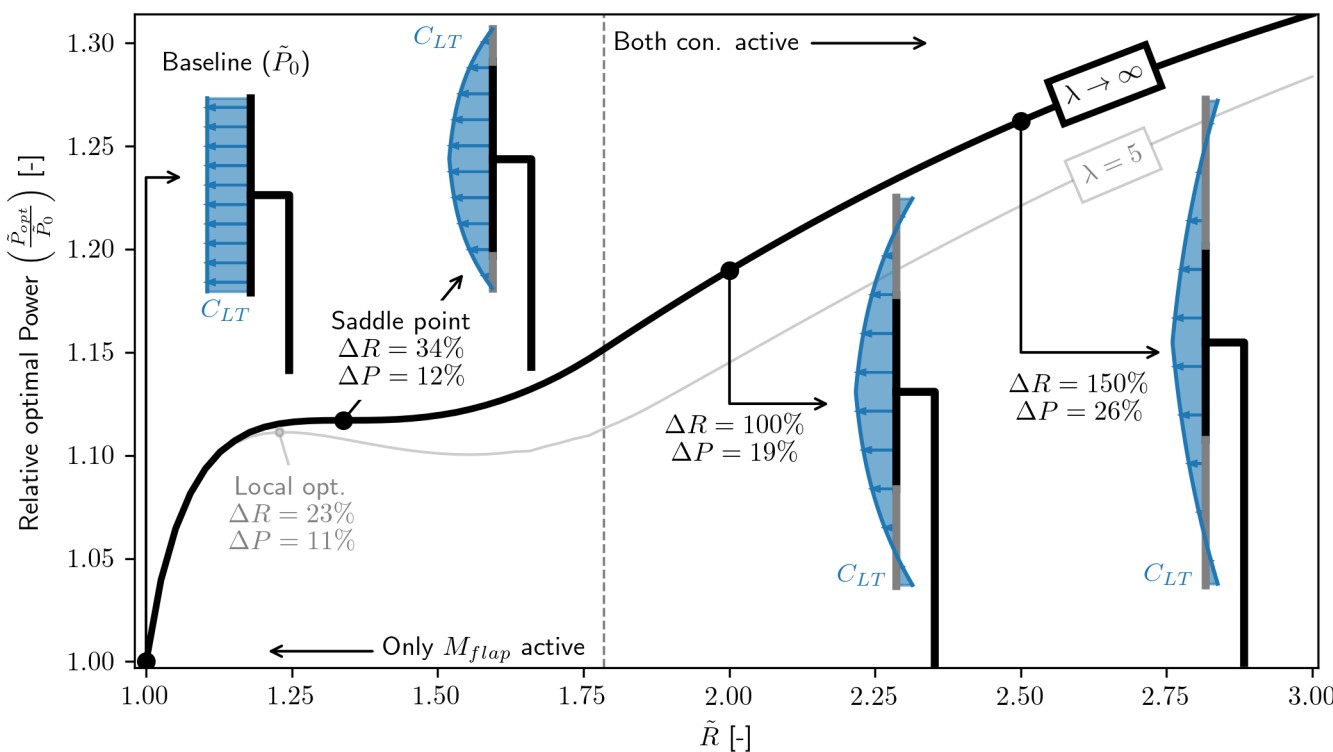

**Figure 5.** Optimal Relative Power ($\tilde{P}_{opt}/\tilde{P}_0$) for increasing radius. The global optimum is at $\tilde{R} \to \infty$. As expected the loading ($C_{LT}$) is seen to taper towards the tip and for large $\tilde{R}$ the loading at the tip becomes negative.

255





For the case of $\lambda \to \infty$ (which means that there are no aerodynamic losses) it is seen that $\tilde{P}_{opt}$ is increasing to a flat plateau (a saddle point) at $\tilde{R} = 1.34, \Delta P = 12\%$. This is a similar result as found by (Jamieson, 2020, p. 810, sec. 3) (they parameterized axial induction and included tip-loss) where the optimal solution is said to be at $\tilde{R} = 1.34$ and $\Delta P = 12\%$. From this analysis (without aerodynamic loss) it is found that this point is a saddle point, but including any aerodynamic loss (or non-optimal loading, like approximate optimal induction) it is found that a local optimum is formed, as it can be seen for the case with $\lambda = 5$ (wake-rotation-loss) where a local optimum is found at $\Delta R = 23\%, \Delta P = 11\%$.

Common to both cases is that the curve is seen to increase again beyond the saddle point/local optimum, and the curves are seen to still increase at $\Delta R = 200\%$. The global optimum is found to have an asymptotic limit as $\tilde{R} \to \infty$, with the optimal power going towards the thrust constraint limit for the case without aerodynamic losses ($\tilde{P}_{opt} \to \tilde{T}_0$, $\Delta P \to 50\%$). Similar behavior is observed for the case with aerodynamic losses. This author observed a similar behaviour using 1D momentum theory but only with a thrust constraint (Loenbaek et al., 2020a, p.163, fig. 6), which was also observed by (Jamieson, 2020, p. 809). To understand why this is also the case when the loading is allowed to vary along the span with thrust and flap-moment constraints, it should be noted that the loading at the tip is negative for large radius increases. The negative loading makes it possible to find a set of load distributions where the flap-moment is zero ($C_{FM} = 0$) but crucially it can still have a positive power ($C_P > 0$). This is all possible while making the thrust loading arbitrary small ($C_T \leftarrow 0$), which in turn means it is always possible to satisfy the constraints for any rotor radius increase while having a positive power ($\tilde{P}_{opt} \to \tilde{T}_0$). Applying a similar argument for the case without a thrust constraint it can be found that the power will grow unbounded ($\tilde{P}_{opt} \to \infty$) since the power coefficient remain finite for increasing rotor radius ($C_{FM} = 0 \to C_P > 0$).

The unbounded behaviour of $P_{opt}$ clearly leads to infeasible designs and for the coming rotor design example a cost function is included to make a realistic rotor design.

### 3.2 Rotor design with cost function

This section will show the result of applying WOwRI for *Power-per-Cost* (PpC) optimization at a single wind speed (assumed to be $\tilde{V} = 1$).

For the rotor design, the aerodynamic losses will be included (i.e. wake-rotation-loss, viscous-loss, tip-loss). To include viscous loss, the glide-ratio ($C_l/C_d$) needs to be given as input, and to get a realistic input for the glide-ratio the DTU 10MW reference turbine (Bak et al., 2013) is used as a basis, in particular the aerodynamic polars as well as the relative airfoil profile thickness distribution along the span. The glide-ratio used for this design can be seen in figure 6 d). Where the polars with relative airfoil thickness $th = [24, 30, 36, 48]\%$ has been used and the design point for the polar is found as it is described in (Bak, 2013, sec. 3.5) and then applying some smoothing, to insure the design will be continues. The $C_l$ and $\alpha$ in figure 6 are used for creating the chord and twist distributions later.

With the glide ratio from figure 6 d) the optimal tip-speed-ratio ($\lambda$) can be found as it is described (Part 1, sec XX). The optimal $\lambda$ and the one used in this section is $\lambda = 8.23$.

A plot of the relative-Power-per-Cost $\left( \frac{\tilde{P}_{opt}}{\tilde{P}_0 f_{cost}} \right)$ for increasing rotor radius can be seen in figure 7. The optimum is found at a radius increase of $\Delta R = 7.9\%$, leading to an increase in Power-per-Cost of $\Delta PPC = 4.2\%$ and a power increase of





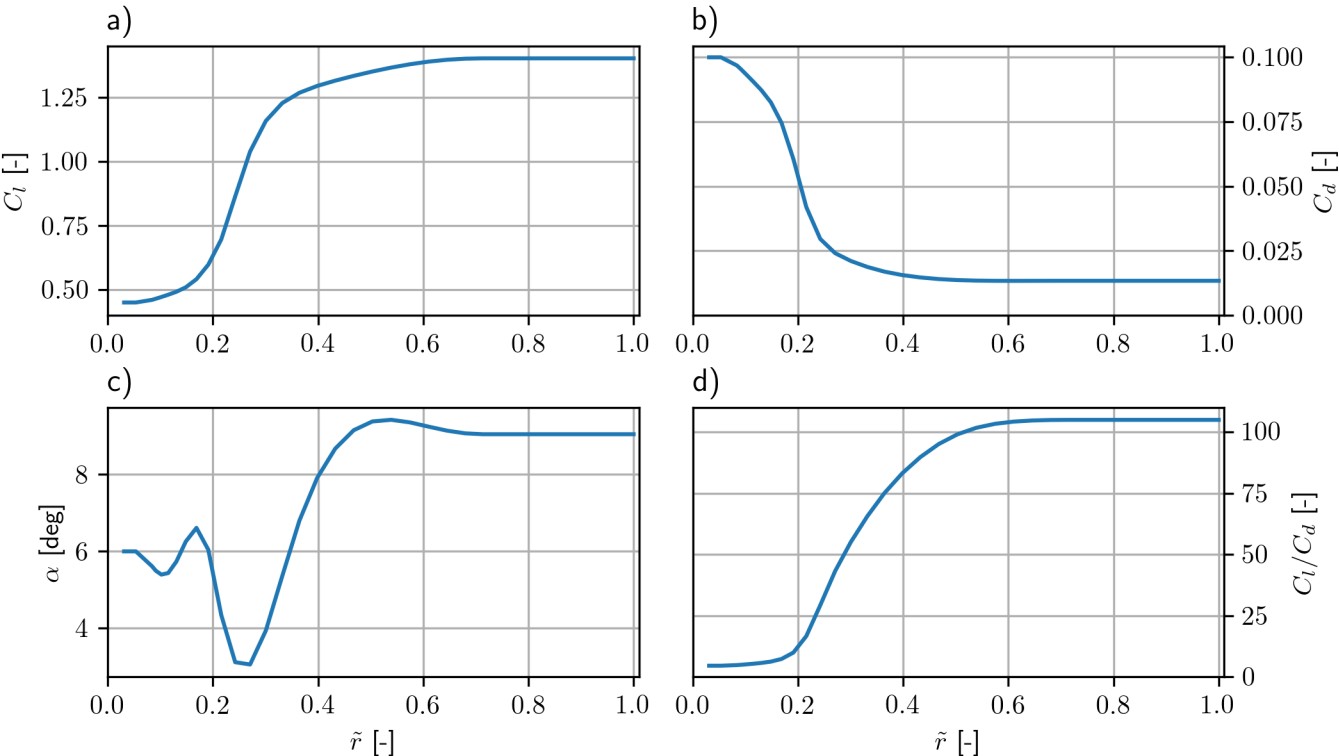

**Figure 6.** Aerodynamic input based on the polars from the 10MW DTU reference turbine. a) Lift-coefficient ($C_l$), b) Drag-coefficient ($C_d$), c) Angle-of-attack ($\alpha$) and d) Glide-ratio ($C_l/C_d$) all as a function of normalized rotor radius ($\tilde{r}$).

$\Delta P = 9.0\%$. From the plot it can also be seen that around $\tilde{R} \approx 1.21$ the impact of increasing the rotor radius leads to a lower power-per-cost relative to the baseline.

A comparison of the loading distribution is shown in figure 8. The plot shows that the loading distribution tapers towards the tip for the optimal design relative to the baseline design. Plot a) shows the thrust-loading-density ($\partial T/\partial r$) per blade (assuming 3 blades) and plot b) the power-density ($\partial P/\partial r$) per blade as a function of the rotor radius ($r$). The solid black line is the value for the PpC-optimized rotor and the difference to the baseline is highlighted with shaded regions, where green indicates a positive impact and red indicates a negative impact. The striking thing to see here is how large the decrease is (the shaded green region) in plot a) and how little impact this lower loading has on the loss of power in plot b) (shaded red region). This has all to do with the fact that operating at maximum $C_{LP}$ a change in $C_{LT}$ will not lead to a proportional change in $C_{LP}$, much like the observation made by this author in (Loenbaek et al., 2020a, p.157 fig. 1) using only 1D momentum theory. Another interesting thing is that it is only the $M_f$ constraint that is active, which means that this PpC-optimized rotor also comes with a lower thrust of $\Delta T = -3.8\%$.

The rotor planform (blade chord and twist) can be found from the loading distribution ($C_{LT}$, figure 7), the lift-coefficient and angle-of-attack ($C_l, \alpha$, figure 6 a), c)), though equation (Loenbaek et al., 2020b, sec. 4.1 eq. 36, 37). A plot of the rotor

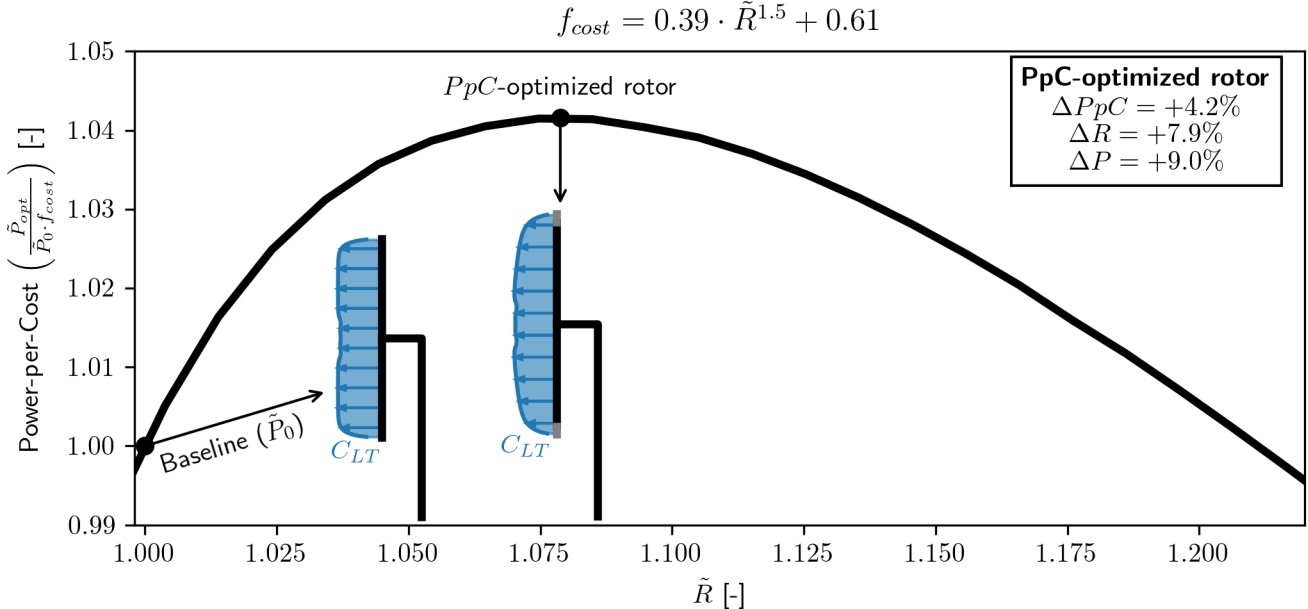

**Figure 7.** relative Power-per-Cost (PpC) vs. radius ($\tilde{R}$). The cost optimized rotor is found to have $\Delta R = +7.9\%$ increase in rotor radius leading to a $\Delta PpC = +4.2\%$ increase.

planform can be seen in figure 9. The figure shows chord and twist for the PpC-optimized rotor, the baseline rotor ($\tilde{R} = 1$)

and DTU 10MW reference turbine. A clear thing to see from these plots is that the optimization did not include a max chord constraint, with the max chord being $\approx 27\mathrm{m}$, which is much larger than the DTU 10MW ref. where the max chord is $6.2\mathrm{m}$. Looking at figure 8, it is seen that the region from $r < 35\mathrm{m}$ has a similar loading as the baseline. Thus the optimization is not exploiting the maximum chord for significant gains and one can safely correct these aberrations after. For $r > 35\mathrm{m}$ the chord is seen to be smaller than the DTU 10MW ref. for both the baseline and the PpC-optimized rotor, with the exception of the

longer blade for the PpC-optimized rotor. Comparing the Baseline and the PpC-optimized rotor, the chord is seen to be the same around $r \approx 60\mathrm{m}$ with the cost-optimized chord being slightly smaller from this point until the tip-loss start to become significant (which is the reason that the chord is going to zero at the tip). The smaller chord is an effect of the tapering $C_{LT}$ for the PpC-optimized rotor. Thus, the lower loading distribution leads to a reduction in the chord. This may have structural implications (i.e. reduced strength and stiffness) that are not accounted for in this optimization.

For the twist (figure 9 b)), the difference between baseline and PpC-optimized rotor is relatively small with an almost constant off-set of $1.5°$ as it can be seen from the $\Delta\theta_{twist}$ plot. The change is fairly small since the flow-angle is approximately $\phi = \tan^{-1} 1/\lambda\tilde{r}$ and the change in $C_{LT}$ only has a small impact.



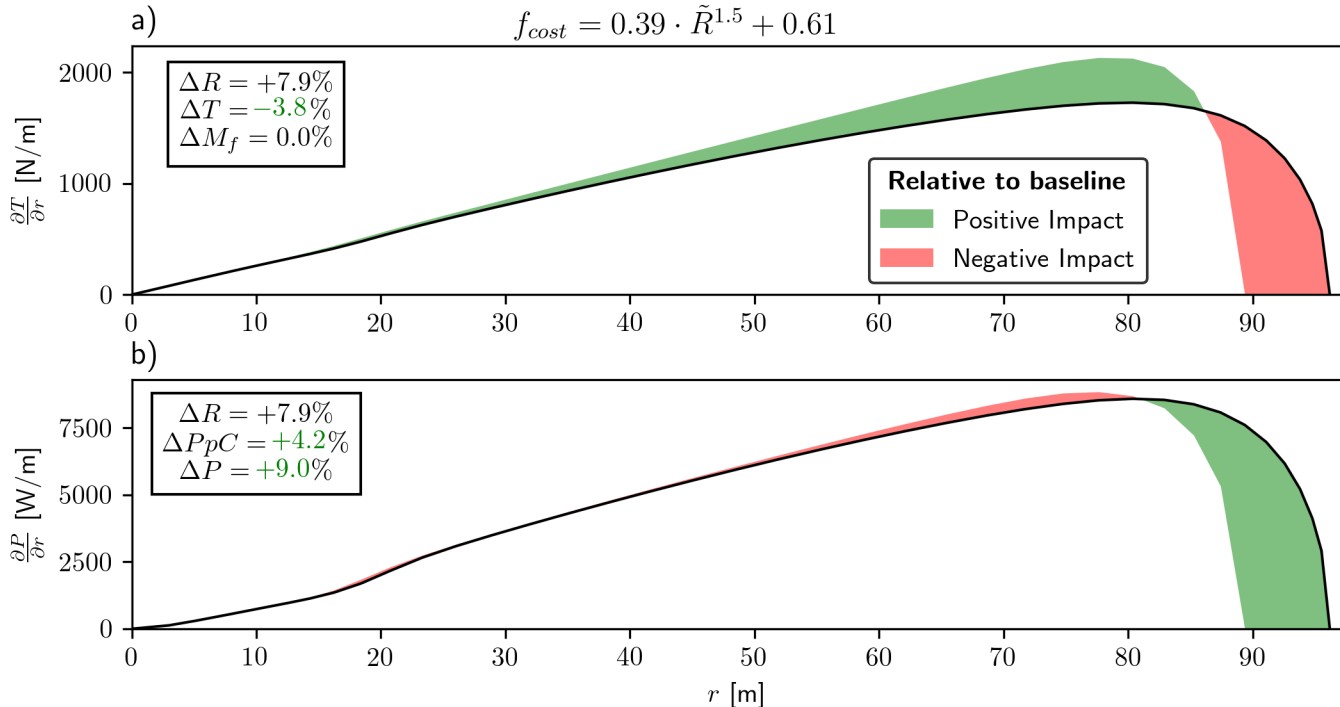

**Figure 8.** a) Thrust-loading-density ($\partial T/\partial r$) b) Power-density ($\partial P/\partial r$) both as a function of rotor radius ($r$). The green shaded regions shows a positive impact relative to the baseline, red regions shows a negative impact. The thing to notice is the significant decrease in the loading (plot a)) and how little impact the lower loading has on the power (plot b)). This is due to the non-linear relationship between $C_{LT}$ and $C_{LP}$.

## 3.3 AEP optimization

In this section the result of solving for the optimal Annual-Energy-Production (AEP) is shown, as it is explained in section 2.3
which resulted in $A\tilde{E}P_{opt}$ (equation 22). The aerodynamic input is the same as for power optimization with no viscous-loss ($C_d/C_l = 0$), no tip-loss, and no wake-rotation-loss ($\lambda \to \infty$), but also including a case with large wake-rotation-loss ($\lambda = 1.5$) to show that a local optimum is formed when aerodynamic loss is added.

When solving the optimization problem in equation 22 the wind speed integration was discretized in 200 steps, which was found to make the discretization error insignificant. The integration is then performed using the trapezoidal rule.

The solution for solving the AEP optimization problem can be seen in figure 10. The AEP-optimization is seen to have similar behaviour as the power-optimization in figure 5 with an initial large slope, followed by a flatter region, then the AEP begins to improve again. The AEP-optimization does not reach a saddle-point or local-maximum for the case of $\lambda \to \infty$, as it was the case for the power-optimization. The slope is always positive. For the case of $\lambda = 1.5$ a local optimum is found, but the formation of this local maximum required a significant amount of aerodynamic loss ($\lambda = 1.5$, which leads to a large

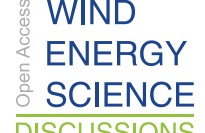

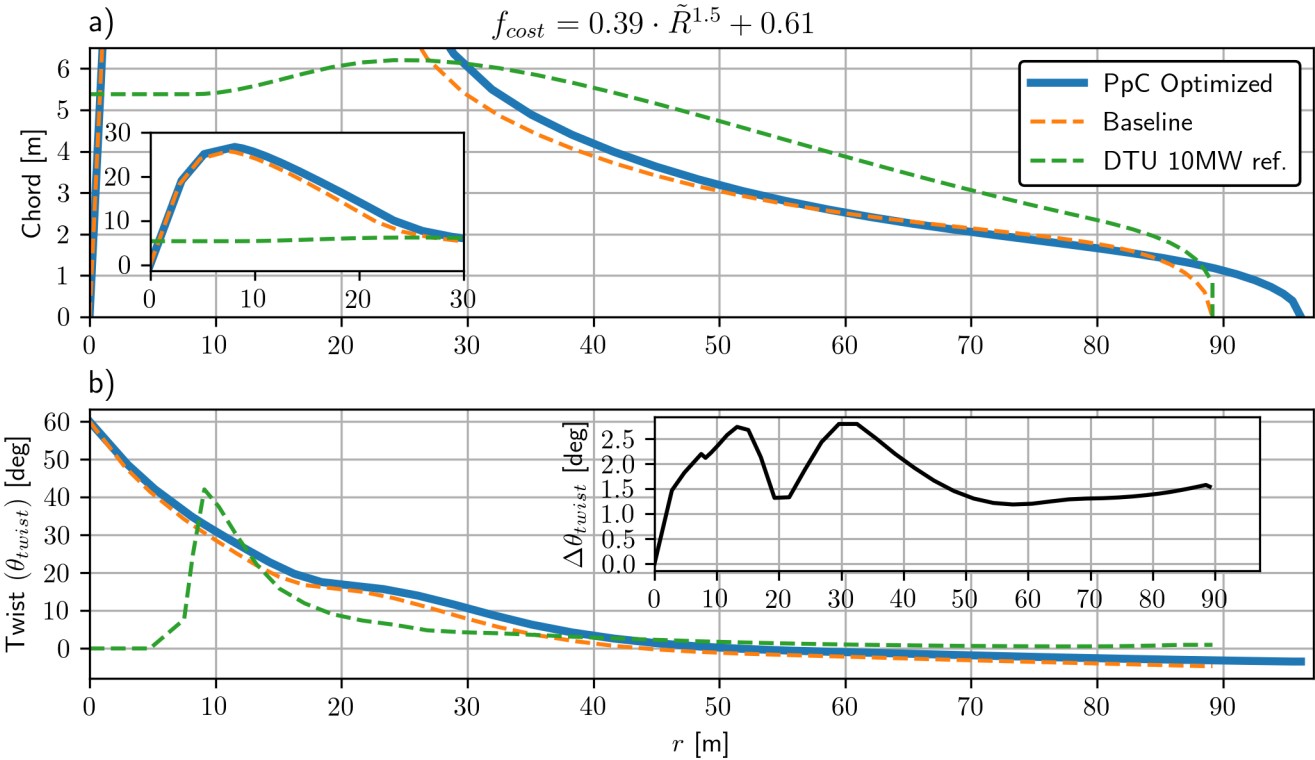

**Figure 9.** a) Blade chord b) Blade twist, both as a function of rotor radius ($r$) for the optimized rotor. In plot a) an insert is added showing the chord from $0 - 30$m. In plot b) an insert is added which shows the difference in twist between the baseline and the PpC-optimized rotor ($\Delta\theta_{twist} = \theta_{PpCopt.} - \theta_{Baseline}$).

wake-rotation-loss) compared to the power-optimization where any aerodynamic loss would lead to the formation of a local-maximum.

     As it was the case for the power-optimization the global optimum for AEP-optimization is found to be a similar asymptotic limit with the optimum as $\tilde{R} \to \infty$ ($\Delta AEP \to 70\%$). This is the case both for $\lambda \to \infty$ and $\lambda = 1.5$. The global optimum for the AEP-optimization tends to a power-curve which almost runs at rated power for all wind speeds, but the maximum power

for a given wind speed is $\tilde{P} = \tilde{T}_0 \tilde{V}$ and for a small region of the power-curve the power will follow this limit before it reaches rated power. This limit is mostly of academic since it is not feasible for practical turbine design, it is not investigated further here.

     Turning to the power- and load-curves for the 4 highlighted point in figure 10. As expected, the $Baseline$ is simply operating at $\max C_P$ until rated power, creating the familiar $\tilde{V}^3$ behaviour for the power and $\tilde{V}^2$ for the loads, where all the peak

loads occur at rated conditions. However, comparing the different optimal solutions along this curve reveals different load profiles than typical modern turbines. Small increases in rotor radius, increase the AEP by reaching rated power earlier. In all the extended rotor cases, the root flap-wise bending moment constraint becomes active before rated conditions are reached.



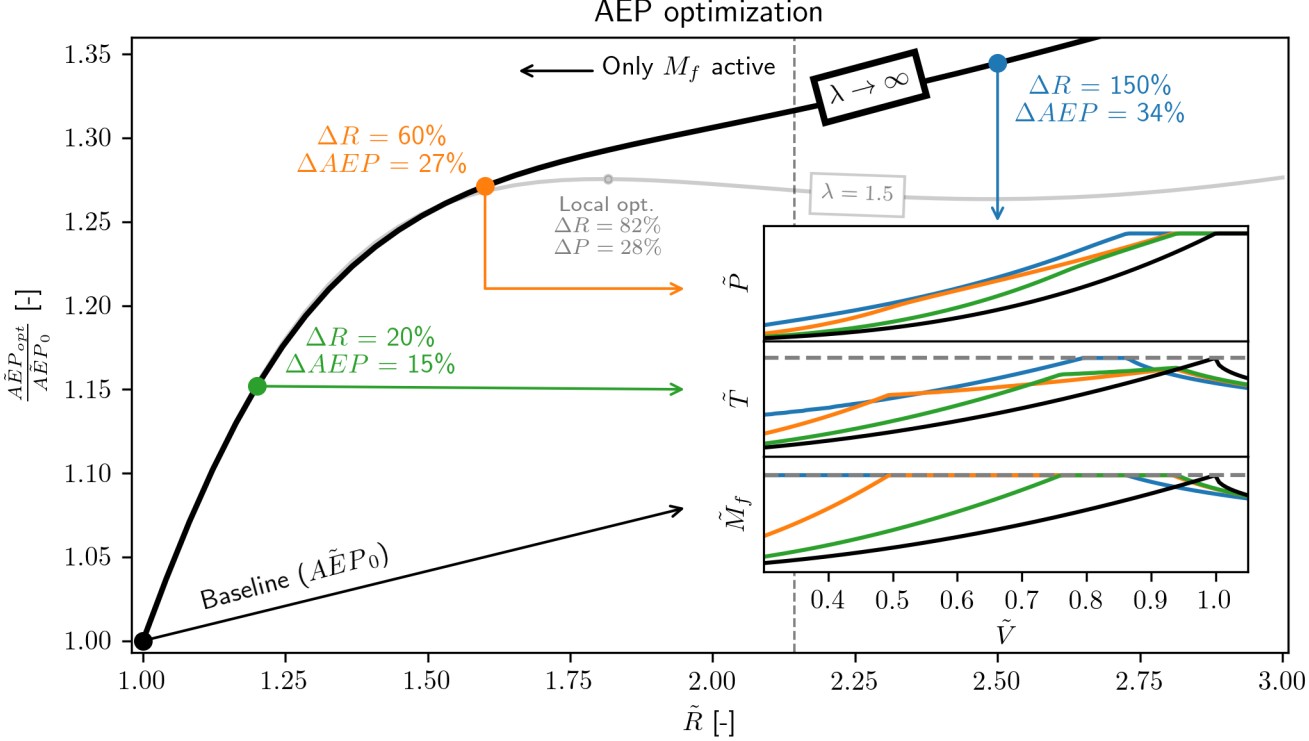

**Figure 10.** Optimal AEP ($A\tilde{E}P_{opt}$) relative to the baseline ($AEP_0$, AEP at $\tilde{R}=1$) vs. relative radius increase ($\tilde{R}$). The vertical dashed line shows the point where the thrust constraint start being active, below this line it is only the flap-moment constraint that is active. The power-, thrust- and flap-moment-curves are show for 4 selected points, which shows how these change for increasing $\tilde{R}$. An additional line shows $A\tilde{E}P_{opt}$ with $\lambda = 1.5$, showing that a local optimum is formed with aerodynamic losses.

Initially, this relaxes the thrust constraint. This bending moment constraint seems to limit the maximum achievable power over a greater range of the power curve. This seems to impose a minimum wind speed that rated power can be achieved, furthermore

increases in AEP must be achieved at lower wind speeds. Finally, for very large rotors, it seems that the moment constraint is active at all wind speeds and the optimization starts to become further constrained by the thrust constraint. In general, the power-curve is found to fall into 3 regimes in terms of wind speed ($\tilde{V}$) which is:

- – Max $C_P$ (no active constraints)

- – Maximizing power with one or more active constraints

– Rated power

These are the same regimes as the author found in (Loenbaek et al., 2020a) using a much simpler model.





### 3.4 Optimal Power Curve with cost function

In this section, the result of solving for the optimal AEP-per-Cost (AEPpC) is presented. At first, the optimal power-curve is presented and at the end, common wind turbine technology is used to see how close it can get to the optimal power curve.

The optimization will use the same aerodynamic input as in section 3.2 (rotor design with cost function), with $\lambda = 8.23$, glide-ratio as in figure 6 d) and including tip-loss.

     AEPpC for increasing values of $\tilde{R}$ can be seen in figure 11, where the AEPpC optimal power-curve is highlighted as well as the baseline. The optimal AEPpC is found to increase by $\Delta AEPpC = 2.9\%$, with a fairly large radius increase of $\Delta R = 17\%$

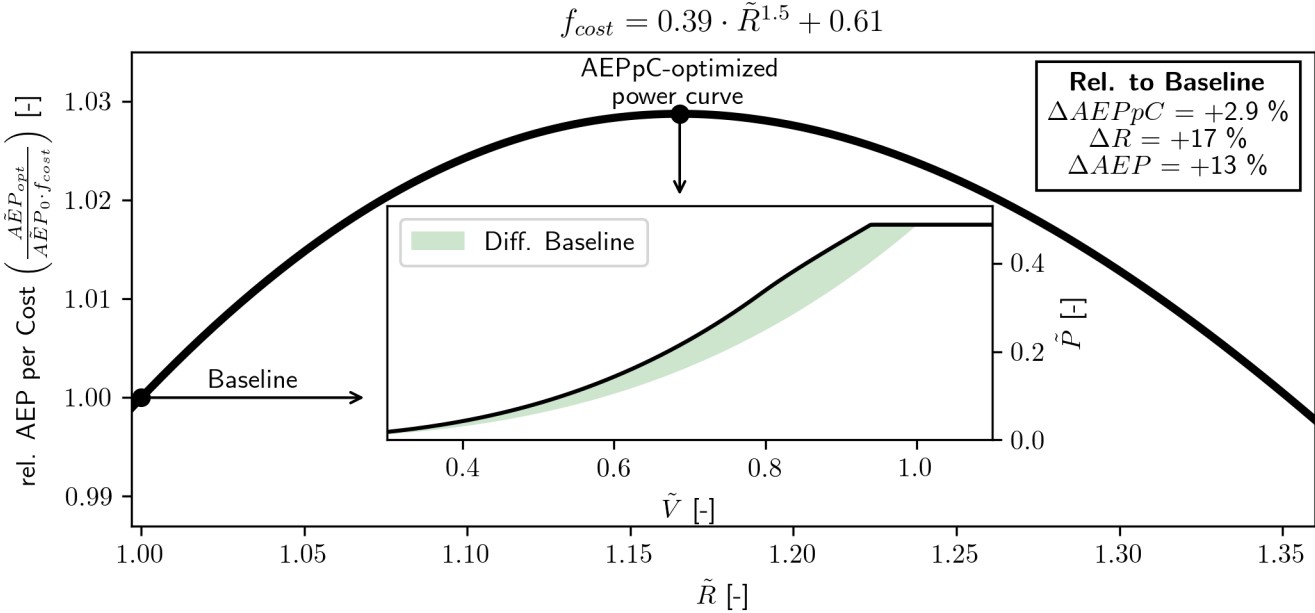

**Figure 11.** Relative AEP-per-Cost (AEPpC) vs. relative radius increase ($\tilde{R}$). The insert shows the cost-optimized power-curve with the shaded region showing the difference to the baseline power-curve. The optimization is seen to reach a cost improvement of $\Delta AEPpC = 2.9\%$ with a radius increase of $\Delta R = 17\%$ as well as an AEP increase of $\Delta AEP = 13\%$.

as well as a fairly large AEP increase of $\Delta AEP = 13\%$. In figure 11 it is also possible to see the power curve as well as the
difference to the baseline. The increase in the power is seen to also increase for increasing $\tilde{V}$ until rated power.

     Figure 12 shows this power curve, along with the loads and loading distribution in greater detail. 3 operational regimes can be seen in figure 12, separated by vertical dashed lines. The optimal power curve is seen to only have an active $M_f$ constraint starting at $\tilde{V} \approx 0.79$ up until rated power at $\tilde{V} \approx 0.94$. In this region, the thrust-curve is seen to change the slope and become linear, but it does not reach the constraint limit (vertical dashed line). The loading distribution ($C_{LT}$) for 4 selected points can
also be seen. Starting from the point just before the $M_f$-constraint becomes active, the loading is the one that maximizes $C_P$ as it has been all the way up until this point. $C_{LT}$ is then seen to progressively taper towards the tip as the wind speed increases.



WIND
ENERGY
SCIENCE
DISCUSSIONS

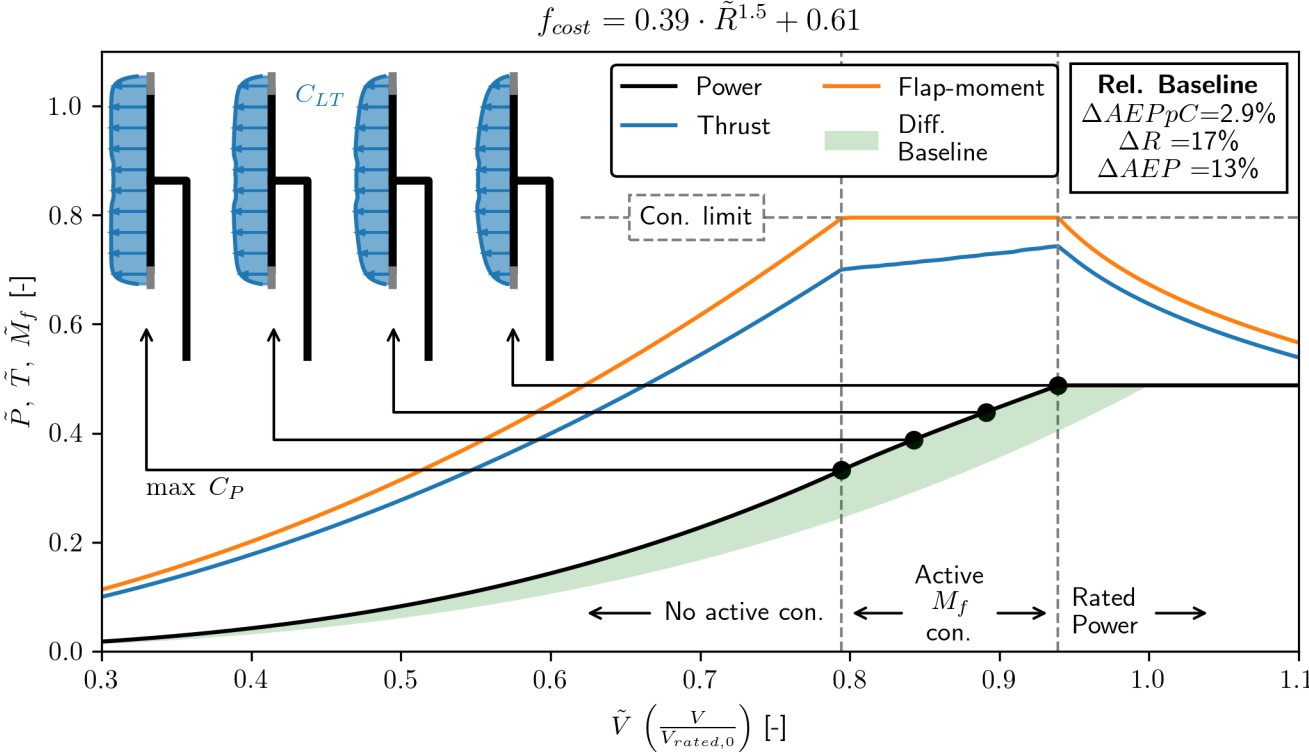

**Figure 12.** Normalized power ($\tilde{P}$), thrust ($\tilde{T}$) and flap-moment ($\tilde{M}_{flap}$) vs. normalized wind-speed ($\tilde{V}$). The transition between the 3 different operational regimes is indicated by the vertical dashed lines. In the region with the active constraint 4 points are selected showing the optimal loading distribution ($C_{LT}$) along the rotor disc.

The presented optimal power curve can not be made into a blade design as was done in section 3.2, since the loading distribution was varied independently at each wind speed. The presented AEPpC-optimization can therefore be seen as the idealized power-curve much like Betz-limit is the idealized maximum power a turbine can achieve. It is therefore not possible,
within the design constraints and aerodynamic modeling, to do any better than this optimal power-curve. In the next section, it is investigated how close it is possible to get to the optimal power-curve using common wind turbine technology.

### 3.4.1   Rotor design with common wind turbine technology

For current utility scale wind turbines, there are two common parameters for altering the loading with changing wind-speed, namely the blade-pitch ($\theta_{pitch}$) and the rotor rotational-speed ($\omega$).
To compute the aerodynamic performance for a turbine where the control parameters are the blade-pitch and rotational-speed the classical *Blade-Element-Momentum* theory (BEM) is well suited. As it is shown in Part 1 (Loenbaek et al., 2020b, sec. 4.2), there is a direct relationship between RIAD and BEM, and the RIAD-BEM is used for the computation of the aerodynamic performance here. BEM requires additional inputs compared to the AEPpC-optimization, namely aerodynamic airfoil polars



at each location along the span as well as a chord and twist along the span. The airfoil polars are taken from the DTU 10MW
ref. turbine, which was the same airfoil polars used to create the glide-ratio input in figure 6. The chord and twist are chosen to
be the loading that maximizes $C_P$ (the loading can be seen in figure 12 as the loading to the left). This is the same chord and
twist as the *Baseline* in figure 9, but with the chord linearly scaled for the radius increase.

In order to directly compare the rotor design with the optimal power-curve the radius increase is assumed to be the same as
for the AEPpC optimized power curve ($\tilde{R} = 1.17$) and the target is then to solve a similar optimization problem as in equation
21 but with the design variables blade-pitch and rotational-speed instead. Mathematically the optimization can be stated as:

$$\int_{\tilde{V}_{CI}}^{\tilde{V}_{CO}} \max_{\boldsymbol{\omega},\boldsymbol{\theta_{pitch}}} \left[ C_P(\boldsymbol{\omega},\boldsymbol{\theta_{pitch}}) \cdot \tilde{R}^2 \tilde{V}^3 \right] f_{wei} d\tilde{V} \tag{26}$$

$$\text{subj.} \quad \left. \begin{array}{ll} C_T(\boldsymbol{\omega},\boldsymbol{\theta_{pitch}}) \cdot \tilde{R}^2 \tilde{V}^2 & \leq \tilde{T}_0 \\ C_{FM}(\boldsymbol{\omega},\boldsymbol{\theta_{pitch}}) \cdot \tilde{R}^3 \tilde{V}^2 & \leq \tilde{M}_{f,0} \\ C_P(\boldsymbol{\omega},\boldsymbol{\theta_{pitch}}) \cdot \tilde{R}^2 \tilde{V}^3 & \leq \tilde{P}_0 \end{array} \right\} \text{for all } \tilde{V}$$

Where the optimization problem is solved in the same manner shown in figure 4, by maximizing the power while observing
the constraints at each wind speed independently.

The result of the optimization can be seen in figure 13, which shows the power- and load-curves as well as the pitch and
rotational-speed traces. The striking thing to notice is how little the difference is between the AEPpC-optimized rotor and the
BEM-optimized rotor. The difference between the two in terms of both AEP and AEPpC is seen to be $0.05\%$ which for all tense
and purposes can be considered an insignificant difference. It means that even with this common wind turbine technology it is
possible to get close to the idealized AEPpC-optimized rotor and it, therefore, seems that the optimization methodology can
almost directly be applied for rotor design. With that said, changes to the optimization problem might lead to the agreement
becoming worse if the region where the constraint is active becomes larger or the limiting constraint is changed. This should
be investigated further.

The optimal BEM rotor design is seen to be achieved through a $\theta_{pitch}$ that is almost linearly in the regime of the active
$M_f$ constraint. $\omega$ is seen to be almost constant after the $M_f$ constraint becomes active. At four points in the regime with
the active $M_f$ constraint the loading is shown, where the difference between the AEPpC-optimized loading and the BEM-
optimized loading is shown with the shaded red area. The difference is seen to get more significant for increasing wind-speeds,
as one might expect. This is also the reason that if the regime of an active constraint is increased the difference between the
AEPpC-optimized- and BEM-optimized-rotor will likely become bigger.



**Figure 13.** Power- and load-curves (top curves) as well as blade-pitch ($\theta_{pitch}$ - left y-axis) and rotor-rotational-speed ($\omega$ - right y-axis) as a function normalized wind-speed ($\tilde{V}$) for the BEM-optimized rotor. Rotor loading at four selected points is shown, with the red region showing the difference to the AEPpC-optimized power-curve load. The difference between the AEPpC- and BEM-optimized-rotor in terms of AEPpC is seen to be insignificant with a difference of 0.05%.





## 4    Conclusion

A novel wind turbine optimization methodology was presented. The crucial assumption that allows for this nested optimization approach is the assumption of radial independence, which is similar to the assumption made in the blade element momentum theory. It allows solving the optimal relationship between the global power- ($C_P$) and load-coefficient ($C_T$, $C_{FM}$) through the use of KKT-multipliers, leaving an optimization problem that can be solved at each radial station independently. It allows for the original optimization problem where the optimization variables are loading distribution $C_{LT}(r)$, to be changed into a

KKT-multipliers for each constraint ($W_0$, $W_1$, etc.).

Applying the optimization methodology for power ($P$) or Annual-Energy-Production (AEP), without a cost-function, leads to the same overall result with the global optimum being unbounded in terms of rotor radius ($\tilde{R}$) with the global optimum being at $\tilde{R} \rightarrow \infty$ with an increase in power or AEP of $\Delta P = 50\%$ or $\Delta P = 70\%$, respectively.

With a simple cost function a Power-per-Cost (PpC) optimization resulted in a Power-per-Cost increase of $\Delta PpC = 4.2\%$

with a radius increase of $\Delta R = 7.9\%$ as well as a power increase of $\Delta P = 9.1\%$. This was obtained while keeping the same flap-moment and reaching a lower thrust of $\Delta T = -3.8\%$. The equivalent for AEP-per-Cost (AEPpC) optimization leads to increased cost-efficiency of $\Delta AEPpC = 2.9\%$ with a radius increase of $\Delta R = 17\%$ and an AEP increase of $\Delta AEP = 13\%$, again with the same, maximum flap-moment, while the maximum thrust is lower than the baseline.



## 5 Nomenclature

### 5.1 Rotor Global variables

Variables that are scalars for the whole rotor. Bold-face variables indicate the variable is a function or vector that changes with wind speed.

| Symbol | Description | Unit |
|---|---|---|
| $\boldsymbol{X}$ | Bold face *global* variables symbolizes function or vector changing with wind-speed ($\tilde{V}$) | - |
| $R$ | Rotor radius | m |
| $T$ | Rotor thrust | N |
| $M_f$ | Rotor root-flap-bending moment | Nm |
| $P$ | Rotor power | W |
| AEP | Annual Energy Production | J |
| $V$ | Free stream wind speed | ms$^{-1}$ |
| $V_{rated}$ | Wind speed at which the rotor reaches rated power | ms$^{-1}$ |
| $\theta_{pitch}$ | Blade pitch angle | deg |
| $\omega$ | Rotor rotational speed | rpm |
| $\tilde{R}$ | Normalized rotor radius ($R/R_0$) | - |
| $C_T$ | Rotor thrust coefficient | - |
| $C_{FM}$ | Rotor flap-moment coefficient | - |
| $C_P$ | Rotor power coefficient | - |
| $\tilde{T}$ | Normalized rotor thrust ($C_T\tilde{R}^2$) | - |
| $\tilde{P}$ | Normalized rotor power ($C_P\tilde{R}^2$) | - |
| $A\tilde{E}P$ | Normalized Annual Energy Production | - |
| $\tilde{f}_{cost}$ | Normalized cost function (only a function of $\tilde{R}$) | - |
| $\tilde{V}$ | Normalized free stream wind speed ($V/V_{rated,0}$) | - |
| PpC | Power-per-Cost ($\tilde{P}/\tilde{f}_{cost}$) | - |
| AEPpC | AEP-per-Cost ($A\tilde{E}P/\tilde{f}_{cost}$) | - |
| $W_i^*$ | KKT-multiplier (Non-scaled Lagrange problem) | - |
| $W_i$ | KKT-multiplier (Optimization variable) | - |
| $\lambda$ | Rotor tip-speed-ratio $\left(\lambda = \frac{\omega R}{V}\right)$ | - |





## 5.2 Rotor Local variables

Variables that are scalars at a given radius location ($r$). Bold-face variables indicates it is a function or vector changing with radius.

| Symbol | Description | Unit |
|---|---|---|
| $\boldsymbol{x}$ | Bold face *local* variables symbolizes a function or vector changing with the local rotor radius ($r$) | - |
| $r$ | Rotor radius variable $[0, R]$ | m |
| $\frac{\partial T}{\partial r}$ | Thrust loading density | $\mathrm{Nm}^{-1}$ |
| $\frac{\partial P}{\partial r}$ | Power loading density | $\mathrm{Wm}^{-1}$ |
| $\tilde{r}$ | Normalized rotor radius variable ($\tilde{r} = \frac{r}{R}$) | - |
| $C_{LT}$ | Local thrust coefficient (normalized $\partial T/\partial r$) | - |
| $C_{LP}$ | Local power coefficient (normalized $\partial P/\partial r$, assumed to be a function of $C_{LT}$) | - |
| $C_l$ | Lift coefficient | - |
| $\frac{C_l}{C_d}$ | Airfoil glide ratio | - |
| $\frac{C_d}{C_l}$ | Inverse airfoil glide ratio | - |
| $\alpha$ | Airfoil angle-of-attack | - |



*Author contributions.* KL came up with the concept and main idea, as well as made the analysis. All author have interpreted the results and made suggestions for improvements. KL prepared the paper and figures with revisions of all co-authors.

*Competing interests.* The authors declare that they have no conflict of interest.

*Acknowledgements.* We would like to thank Innovation Fund Denmark for funding part of the industrial PhD project which this article is a part of.

We would like to thank all employees at the former Suzlon Blade Sciences Center (Vejle, Denmark) for giving valuable feedback in the initial phase of the development.

We would like to thank Antariksh Dicholkar from DTU Risø for many good discussion and inputs for regarding the work.



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
