# Peer review of "A Method for Preliminary Rotor Design - Part 2: Wind Turbine Rotor Optimization with Radial Independence"

_Wind Energy Science, 2020_

## Referee Comment (RC1) · Anonymous Referee #1 · 3 Nov 2020

In the paper, the potentiality for load-constrained optimization of horizontal-axis wind turbines (HAWTs) of the methodology developed by the authors in the part 1 of the paper (RIAD) is presented. Based on the same theory of Blade Element Momentum (BEM), this approach allows nonetheless to re-parametrize the design problem in terms of blade spanwise load and power distribution, leading to a more physically sound interpretation and faster convergence of the optimization process when the turbine maximum loads are imposed as a constraint. The potentiality of the proposed methodology is first demonstrated by maximizing the power and Annual Energy Production (AEP) of a test turbine in case of unbounded rotor dimension. The same process is then repeated by bounding the machine size via an ad hoc cost function, showing promising

results.

The reviewer believes that the topic and the activity are very interesting, innovative and worthy of investigation. The adopted methodology is rigorous and clearly detailed throughout the whole paper, which is very well presented. Based on the aforementioned comments, the publication of the paper in the present form is strongly recommended.

Some technical corrections: Line 46: replace "part 2 of 2 part paper" with "part 2 of two-part paper"; Line 34: the name "Chaviaropoulos" is repeated twice;

---

## Referee Comment (RC2) · Anonymous Referee #2 · 23 Nov 2020

This paper is the second of a two-part series on Rotor Design. This paper deals with the application of the radial independent actuator disk model (RAID) to maximize power or Annual-Energy-Production for a given thrust and blade-root-flap-moment. The model includes a simple cost function which makes a difference in the solution space. The result is an improvement in performance that is worth the effort of using this model. Defining the optimization as "nested" leading to the overall global optimization. The result is a simplification of the process hopefully without loosing needed information. The optimization section is straight forward and presented in a logical fashion with sufficient reference to Part 1 (on page 7 you do have and eq. XX that needs attention). Optimizing each section allows the performance to be tailored to the conditions at each radial

station. It is interesting as the radius increases how the power distribution changes over the rotor. As pointed out in the paper, a max chord would be desirable as the resulting 27 m chord is not reasonable. The new addition is the cost function which gives a more accurate indication of the rotor performance. This would be a useful tool to use when comparing designs. I would hope that some additional discussion on why one should use this model as compared with current design methods would be included.
* * *

---

## Editor Comment (EC1) · Alessandro Bianchini (Editor) · 2 Feb 2021

Dear authors, reviewers' comments are in since a while. Please try to address them at your earliest convenience. Best regards

---

## Editor Comment (EC2) · Alessandro Bianchini (Editor) · 9 Mar 2021

Dear Authors, I have reviewed your responses to Reviewers' comments. Based on them, I encourage you to resubmit a revised version of your study that will be reconsidered for publication.

---

## Author Comment (AC1) · 9 Mar 2021

Dear Referee,

Thank you for reviewing our manuscript and reading it carefully. You have given a very nice and concise summary of the manuscript. We implemented/corrected your editorial comments.

---

## Author Comment (AC2) · 9 Mar 2021

Dear Referee,

Thank you for reviewing our manuscript and reading it carefully.

*on page 7 you do have and eq. XX that needs attention*
The correct equation has been added (eq. 26).

*a max chord would be desirable*
Applying a max chord is relatively easy to implement as there is a direct relationship between CLT and chord once Cl is given. Limiting the chord is therefore a matter of

finding the maximum CLT for a given radius and then limit it there. It was omitted here to avoid the manuscript of becoming unnecessarily complicated. A comment has been added for the first part of the "Result and discussion" (section 3) which briefly discus the possibility of adding more or other constraints.

*I would hope that some additional discussion on why one should use this model as compared with current design methods would be included.*
A comment has been added in the first part of "Result and discussion" (section 3) where the advantages of using the methodology is mentioned. The main advantages is the speed at which the methodology can obtain results, which makes it possible to search large part of the wind turbine rotor design space - which would be computationally expensive with most aeroelastic solvers.

---

## Author Response (AR1)

Dear Reviewers,

Thank you for reading and reviewing our manuscript carefully.

Based on your comments we have made the following main changes (see the diff document for the changes):

- (p. 2 line 34, line 46) Removed the repeated name Chaviaropoulos and changed from 2 part to two-part
- (p. 7 line 162) Inserted the correct equation number
- (p. 12 line 250) Added a paragraph which explains the advantages of using this methodology as well as the possibility for adding more or other constraints.